# Exploring the Relationship between Perceived Community Support and Psychological Well-Being of Tourist Destinations Residents

**DOI:** 10.3390/ijerph192114553

**Published:** 2022-11-06

**Authors:** Haihong Wang, Hongxia Sha, Litong Liu, Hengwei Zhao

**Affiliations:** 1Department of Tourism Management, Business School, Liaoning University, Shenyang 110036, China; 2Department of International Economy and Trade, International School, Jinan University, Guangzhou 511486, China

**Keywords:** organizational support theory, perceived community support, psychological well-being, resident–tourist interaction, psychological resilience

## Abstract

To explore the relationship between community support in tourist destinations and residents’ psychological well-being in the post-COVID-19 pandemic period, this study adopts the questionnaire survey method and draws the following conclusions by constructing a structural equation model: (1) perceived community support is very helpful for the psychological well-being of residents, (2) psychological resilience significantly mediates the relationship between perceived community support and residents’ psychological well-being, (3) the resident–tourist interaction mediates the relationship between perceived community support and residents’ psychological well-being, and (4) the resident–tourist interaction and psychological resilience play an ordered chain-mediating role between perceived community support and residents’ psychological well-being. These findings not only fill the gap in tourism research regarding destination-based community support studies but also provide a theoretical basis for maintaining residents’ psychological well-being in a given destination in the context of the COVID-19 pandemic. To a certain extent, improving residents’ well-being is helpful for promoting the healthy and sustainable development of tourism activities and realizing a “win-win” situation in which tourist destinations develop economically while promoting their residents’ living standards.

## 1. Introduction

Since the outbreak of the COVID-19 pandemic, people’s lifestyles have been forced to change, and nearly 20% of the world’s population is concerned about their psychological well-being to some degree [1]. Moreover, many tourism-related businesses, groups, and organizations have faced a sharp decline in revenue, closure, or even bankruptcy due to the unprecedented blow to the tourism industry caused by the COVID-19 pandemic. In this process, the resident local tourism practitioners’ psychological well-being has become a concern. On the one hand, the perceived support and assistance received from the community of a tourism destination is beneficial to the residents’ psychological well-being; on the other hand, the ability of a tourism destination’s residents to recover from stress and welcome tourists again may be closely related to their psychological well-being.

The current COVID-19 pandemic has not been completely eradicated, and previous studies have mostly focused on healthcare workers’ psychological well-being rather than that of the local community. The literature suggests that efforts can be made to improve the safety of working conditions, the level of training, and compensation [2,3,4]. Fukuti et al. (2020) proposed that a dedicated research unit be established to monitor healthcare workers’ psychological well-being so that they feel cared for and supported by the local community and relevant organizations [5]. However, Sun et al. (2021), in their study of the stress brought to CMHW by the COVID-19 pandemic, found that suspected patients who felt isolated by their community showed high levels of depression and anxiety [6]. This also implies the need for community support and research on the residents in particular.

Based on organizational support theory, this study focuses on the association between residents’ perceived community support and psychological well-being from the residents’ perspective and incorporates two variables—psychological resilience and the resident–tourist interaction—as mediating factors. This study focuses on residents’ psychological well-being and its influencing factors in tourist destinations that were affected by the COVID-19 pandemic, emphasizes the importance of community support, and proposes that perceived community support can improve residents’ psychological well-being by enhancing their psychological resilience and positive attitudes toward interacting with tourists.

## 2. Literature Review and Hypothesis

### 2.1. Organizational Support Theory

Eisenberger et al. (1986) integrated the principle of compensation and the idea of organizational anthropomorphism and proposed the concept of organizational support theory (OST) [7]. Previously, researchers have focused too much attention on employees’ commitment to an organization and have neglected the importance of “top-down” commitment. The emergence of organizational support theory has forced leaders’ thinking to shift such that it emphasizes that organizations should focus primarily on their responsibilities and obligations to their employees. Perceived organizational support (POS) is the core concept of organizational support theory. It can be summarized in the following statement: “the organization values my contribution and considers my well-being” [7]. A high perception of organizational support motivates employees’ work behaviors and attitudes because it shows that within the company the organization values employees’ efforts and affords them humane and fair conditions and that high input can be exchanged for high returns [7].

Studies have identified organizational support as being multidimensional and theoretical frameworks have been developed to examine its impact. Thompson and Jahn (2003) argued that the sense of organizational support consists in tangible support at the material level and intangible support at the spiritual-emotional level [8]. Following Eisenberger, scholars proposed a functional model of social support in which it is important to provide support in forms such as equipment, information, and tools to those in need to help them solve the problems they encounter in their work or life [9,10]. Muse and Stamper (2007) argued that perceived community support can be divided into emotional factors related to the community’s own social relationships and work-related factors [11]. Chong et al. (2001) further divided perceived organizational support into top management, middle management, first-line direct supervisors, employees, and executive support according to the position of the leader [12].

This study argues that organizational support theory can explain the changes in residents’ psychological well-being when they perceive support from the community. This stems from the following points: first, similar to corporate teams, residents of tourism destination-based communities are independent individuals within the larger organization of the community; second, their roles are similar, in that communities are formed by residents and serve residents, and the functions they have, such as management, service, education, and supervision, ensure that each resident in the community is supported in their reasonable demands and satisfies their aspirations for a better life; and third, communities also participate in leadership, organization, coordination, control, and other organizational behaviors. Inevitably, these residents must interact directly with different types of tourists in their social networks, and in the event of more serious consequences, such as consumer fraud, the community needs to step in and coordinate the situation.

### 2.2. Perceived Community Support and Psychological Well-Being

Perceived community support is the subjective feeling of being in a community setting in which individuals feel helped and cared for by a social group [13,14]. It is also considered a desirable outcome when employees perceive organizational support and behave in a way that the organization views favorably [13]. In a sense, tourism destinations represent a large and complex organization in the tourism industry [15]. Residents of tourism destinations provide various tourism products and services to tourists; from this perspective, the residents are also “employees” of the destination-as-organization [16]. Therefore, perceived community support is a unique and important presence in perceived organizational support, and the community environment in tourism destinations is also applicable to the theory of organizational support, which reflects residents’ subjective perceptions of the care they receive from the community in all aspects of their lives and careers [17]. 

The concept of psychological well-being is derived from the specific academic subdivision of well-being into subjective and psychological well-being. Subjective well-being refers to an individual’s feelings about their current quality of life using some subjective evaluation criteria [18]. In contrast, Waterman (1993) argued that psychological well-being focuses on the realization of human potential and self-worth [19]. Currently, indicators such as self-acceptance, personal growth, life purpose, and self-actualization are commonly used to study psychological well-being [20]. From the perspective of community residents in tourism destinations, this study defines psychological well-being as a state without serious mental illnesses, the possession of a high degree of satisfaction with one’s overall living environment, and the engagement in positive spiritual pursuits, as well as the good adaptability and problem-solving abilities of the residents living in tourism destinations.

In a study of the relationship between community support and psychological well-being, Zamora et al. (2020) revealed that informal community support is crucial for older adults, and that the level of social and community support will determine the rate at which independence is lost [21]. Jaye et al. (2022) pointed out the importance of health and well-being to the residents of small rural communities in New Zealand through a survey, wherein the more remote the residents are, the more they value the assets of rural communities and the contribution their infrastructure makes to their well-being and health [22]. Moreover, psychological well-being has been further explored in the previous literature in terms of various dimensions such as job satisfaction, stress, commitment, and quality of life. For example, Maan et al. (2020) confirmed that perceived organizational support is associated with the satisfaction and performance levels of employees in companies [23]. Pahlevan et al. (2022), taking medical personnel an example, confirmed a positive correlation between nurses’ perceptions of organizational support and job satisfaction and a lower turnover rate [24]. Chan et al. (2022) considered older adults and younger adults as control groups, and the results showed that in the context of the COVID-19 pandemic, older adults showed better psychological well-being than younger adults [25]. Harrison et al. (2022) investigated the degree to which pregnant women with lower levels of perceived support would experience more negative thought patterns such as depression and anxiety disorders under the stress of the new crown pneumonia pandemic in the UK, which exacerbated the psychological well-being-related challenges faced by pregnant women [26]. In addition, Su et al. (2019) confirmed that both perceived community support and community identity contribute to residents’ quality of life. In turn, quality of life is inextricably linked to psychological well-being [17]. Through an online survey of adult women undergoing infertility treatment, Shin et al. (2021) revealed that perceived social support can positively influence the quality of reproductive life, which is improved by healthcare providers and nurses remaining in close contact with the women during treatment in this population [27].

The existing research on “perceived community support and psychological well-being” has focused primarily on employees, patients, and older adults as the main groups. The literature on the subject has frequently used stress, emotions, illness, and the work environment as independent variables to study their effects on psychological well-being [28,29]. However, little attention has been paid to problems related to psychological well-being in ordinary community residents apart from special groups of people, and the research scope is less likely to involve tourism in general and tourism destination communities in particular. Considering that this industry involves communities in which residents perceived that strong community support may contribute to the recovery of residents’ psychological well-being, this study proposes the following hypothesis:

**Hypothesis** **1.***Perceived community support positively contributes to the psychological well-being of the residents of tourism destinations*.

### 2.3. The Mediating Role of Psychological Resilience and Resident–Tourist Interaction

Kathryn and Jonathan (2003) defined resilience as the important ability or quality of a person to withstand the stresses faced in unexpected situations and be able to adjust quickly [30]. The COVID-19 pandemic is such a unique case, for which resilience theory can provide some theoretical guidance for the construction and development of current tourism destinations. For example, Traskevich and Fontanari (2021) constructed a conceptual, integrated model of tourism destination resilience in the post-pandemic era, which confirmed that the concept of destination resilience is beneficial for tourism’s attractiveness and competitiveness through a survey of more than 1000 tour operators in Germany [31]. Ngoc et al. (2021) conducted a study in the form of interviews of employees and residents of hotel enterprises in Vietnam and suggested that implementing valuable human resource resilience-building measures could help maintain labor force engagement in the tourism industry and enhance organizational resilience [32]. Psychological resilience belongs to the study of positive psychology. The American Psychological Association (2016) also considers psychological resilience as the thoughts and behaviors that individuals learn and develop when recovering from and adapting to adversity, threats, or stress [33].

In the relationship between perceived community support and psychological resilience, Liang (2022) confirmed that social support has a positive direct effect on psychological resilience [34]. Xu et al. (2022) investigated medical residents and found a significant mediating effect of psychological resilience between social support and coping styles [35]. Kong et al. (2021) found that psychological resilience partially mediates the relationship between social support and health-related quality of life through a survey of older-adult immigrants [36]. Park et al. (2022), through an analysis of previous survey data, showed that psychological resilience is an effective coping resource for communities facing stress from crises, thus facilitating the recovery of disaster victims [37]. 

In the relationship between psychological resilience and psychological well-being, it is generally accepted that psychological resilience is related to psychological well-being (i.e., those with higher levels of psychological resilience will have fewer problems related to psychological well-being) [38]. Yang et al. (2020) concluded that patients’ psychological resilience is also related to their psychological well-being, quality of life, and lack of illness, and that patients with high psychological resilience will experience greater post-traumatic growth in the face of a given illness [34]. Satici (2016) found through a survey of Turkish university students that psychological resilience and hope have a positive relationship with well-being and that hope mediates the relationship between the two [39]. Applied to the tourism destination setting, the residents in these areas are still in a state of self-recovery after experiencing the tremendous economic and mental stress brought about by the COVID-19 pandemic, in which case their level of psychological resilience will affect their psychological well-being. Thus, this study presents the following hypothesis (2).

**Hypothesis** **2.***The relationship between perceived community support and residents’ psychological well-being is positively mediated by psychological resilience*.

The “Resident-tourist interaction” refers to McNaughton’s (2006) “host and guest” perspective with respect to interpreting the relationship between residents and tourists, which considers the residents of tourism destinations, groups engaged in the tourism business, etc., as “hosts” and the tourists who engage in tours, sightseeing, consumption, etc., as “guests” [40,41]. Nunkoo (2016) used social exchange theory to study the attitudes of destinations’ residents towards tourism, showing that the value of the economic, social, and cultural elements of the resident–tourist exchange process influence the way residents perceive tourism development and determine the degree of their acceptance of it [42]. Sutton (1967) argued that this interaction might provide opportunities for communication and reinforce the impulse for residents to develop their marketable skills [43]. From the perspective of community residents, this study defines the “resident-tourist interaction” as the sum of material, informational, and emotional exchanges between community residents engaged in tourism activities and visiting tourists in the tourism destination environment. In recent studies, it has also been found that contact between residents of tourist destinations and tourists helps to improve intercultural relations [44,45]. In addition, if residents hold positive attitudes toward tourists during their interactions, this can enhance community support for local tourism development [46,47]. 

According to organizational support theory, employees should be treated fairly in material and emotional terms so that they can feel supported and cared for by the organization, which will also prompt them to give feedback on organizational support. Al-Omar et al. (2019) identified a strong relationship between pharmacists’ perceptions of organizational support and their level of engagement at work, and that those employees who perceive that they feel organizational support are more likely to behave in an engaged manner [48]. Chiang et al. (2012) also confirmed the positive impact of perceived organizational support on organizational citizenship behavior through a survey of hotel employees [49]. For residents, who are stakeholders in the destination, this feedback can be expressed as their willingness to interact with tourists in a friendly manner after perceiving support from the community, increasing tourists’ goodwill towards the destination, and thus responding to community support. This study argues that the resident–tourist interaction is likely to represent positive feedback for perceived community support.

At the same time, in the development of tourism, local residents can provide tourism services and show local customs to tourists in either a direct or an indirect way [41], and this social interaction and associational enhancement are key drivers of well-being [50]. Thus, resident–tourist interactions may affect residents’ psychological well-being. Yu and Lee (2014) argued that positive interactions between tourists and local residents are conducive to building memorable tourism experiences [51]. Richard (2014) indicated that in addition to enhanced tourism experiences for tourists, the hosts’ own attitudes, perceptions, and behaviors are also affected [52]. Ye et al. (2020) also confirmed that the participation of a destination’s residents and tourists in value co-creation has a positive impact on tourists’ subjective well-being [53]. The current research confirms that the quality of the resident–tourist interaction plays a significant mediating role in the satisfaction and loyalty of tourists towards a destination [54,55]. Stylidis’ study (2022) found a positive impact of the resident–tourist interaction on the image of tourism destinations [56]. Woosnam (2022) delved more deeply into these findings by showing that the quality of this interaction not only positively affects the destination’s image but further shapes satisfaction and loyalty [57]. In the previous discussion of psychological well-being, satisfaction is also closely related to residents’ psychological well-being. Thus, we present Hypothesis 3:

**Hypothesis** **3.***The relationship between perceived community support and residents’ psychological well-being is positively mediated by the resident–tourist interaction*.

Based on the above analysis, it can be seen that perceived community support can be associated with psychological well-being not only through psychological resilience but also through resident–tourist interactions. However, it is worthwhile to explore whether there is also an association between psychological resilience and resident–tourist interactions. Building on previous research that shows psychological resilience increases employees’ well-being, job engagement, and organizational commitment [58,59], Kim et al. (2004) argued that psychological resilience in turn promotes employees’ service orientation, including customer focus and customer service under stress [60]. Prayag (2020) investigated the influencing mechanisms between multiple types of resilience through a survey of tourism business owners and employees who experienced an earthquake disaster, and the findings not only showed a significantly positive relationship between psychological resilience and employees’ resilience, but also indicated that employees’ resilience contributes to the life satisfaction and organizational resilience of tourism business operators and that the dynamic relationship between the high-psychological resilience characteristics of residents and the resident–tourist interaction is worth further study [61].

Combined with the previous analyses, we can hypothesize that residents who perceive more community support contribute to psychological resilience, and residents with higher psychological resilience enhance their interactive behaviors with visitors as hosts; as a result, their psychological well-being improves in friendly interactions. Therefore, this study proposes Hypothesis 4:

**Hypothesis** **4.***The relationship between perceived community support and residents’ psychological well-being is sequentially and positively mediated by psychological resilience and resident–tourist interactions*.

Figure 1 shows a conceptual model of this study to illustrate the relationship between the four hypotheses. According to this conceptual model, the psychological well-being of residents of tourist destinations will be associated with three aspects: the degree of perceived community support, psychological resilience, and resident–tourist interactions.

## 3. Research Methodology

### 3.1. Sample and Sampling

A popular village destination in Zhejiang Province, China, named “Gu Yan Hua Xiang” (hereafter referred to as Guyan) was selected as the study location. Relying on the superior local natural resources and the correct guidance of the government, Guyan has invested in the construction of the first batch of characteristic towns in China to create a multifunctional demonstration area integrating a sketching, creation, and oil-painting production base, as well as a leisure and vacation center. In 2020, when the COVID-19 pandemic was raging, Guyan strictly followed the relevant regulations to suspend its business measures. At present, even if the scenic spot resumes operation, it always adheres to regular epidemic prevention and control measures and limits the flow of visitors. Therefore, taking the residents within the region as an example is representative of examining the interrelationship between perceived community support and psychological well-being.

The study was divided into two phases. In the pre-research phase, the research team conducted a pilot study with 20 tourism management students. The students were asked to consider their hometown as a tourist destination from the residents’ perspective and to assess each issue for themselves. Some of the participants in this study were confused about the concept of the “Gu Yan Hua Xiang Community”. Based on this feedback, the final questionnaire was clarified to refer to “the Guyan scenic area or tourist site, not specifically to the street or community where it is located”.

In April 2022, based on the reopening of Guyan after the COVID-19 pandemic, residents engaged in the tourism industry returned to their jobs and started their businesses; so, there were more subjects who fit this interview. In order to conduct the research more smoothly, the research team hired a local guide. The research team was divided into small teams of 2 researchers to cover these areas. Respondents completed the questionnaire through an online questionnaire platform. To gain insight into residents’ views and to avoid confusion, at least one researcher was available during the process to objectively answer participants’ questions. The research team ensured the anonymity and confidentiality of all responses. A total of 220 questionnaires were distributed in this study, and 207 valid questionnaires were obtained after eliminating all questionnaires with identical answers and those with less than 180 s of answer time for an effective response rate of 94.1%. The entire process lasted for 14 days, and the survey was conducted during lunch breaks and evening closing times to avoid disturbing normal business.

### 3.2. Variables and Tools

The variables of perceived community support, resident–tourist interaction, psychological resilience, and psychological well-being were measured in this study using a seven-point Likert scale. For data analysis, descriptive statistics and correlation analysis of the data were conducted mainly using SPSS (version 22.0, IBM Corp., Armonk, NY, USA) software; reliability was judged by standardized Cronbach coefficient values; validation factor analysis was conducted using AMOS 24 software; and bootstrap testing of model paths was employed to perform bias correction, setting 5000 replicate sampling times and 95% confidence intervals.

(1) Perceived community support. Perceived community support was measured using the Perceived Organizational Support (POS) scale developed by Settoon et al. (1996) [62]. The scale in this study consisted of four main items and the measurement items were assessed using a seven-point Likert scale (1 = strongly disagree; 7 = strongly agree), which had a Cronbach’s alpha coefficient of 0.882 and showed good internal consistency and reliability.

(2) Resident–tourist interaction. Community resident–visitor interactions were measured using the Visitor–Resident Interaction Quality (VRIQ) scale developed by Teye et al. (2002) [63]. It consists of five main items, and the measurement questions were assessed using a seven-point Likert scale (1 = strongly disagree; 7 = strongly agree), which has a Cronbach’s alpha coefficient of 0.860 and showed good internal consistency and reliability.

(3) Psychological resilience. The CD-RISC scale developed by Stein (2007) was used [64]. Five question items were included. The measurement items were assessed using a seven-point Likert scale (1 = strongly disagree; 7 = strongly agree), which has a Cronbach’s alpha coefficient of 0.849 and good internal consistency and reliability.

(4) Psychological well-being. The Psychological well-being (PSB) scale of Ryff and Keyes (1995) was mainly used to measure this variable [20], and four main question items were screened in the current study (as shown in the table). The measurement items were assessed using a seven-point Likert scale (1 = strongly disagree; 7 = strongly agree), which has a Cronbach’s alpha coefficient of 0.772 and good internal consistency and reliability.

## 4. Data Analysis

### 4.1. Preliminary Analysis

Among the 207 valid samples, the proportion of female respondents (58.0%) was higher than that of male respondents (42.0%); the age groups of respondents were concentrated between 31–40 years old (31.9%), 41–50 years old (34.3%), and 51–60 years old (27.5%), with most of them being young and middle-aged. Among them, 98.6% of them were long-term residents of Guyan whose education level was moderately low (62.4% in junior or senior high school). Only 37.7% of them had a bachelor’s degree or above, while their monthly incomes were mostly between 2001 and 8000 yuan (98.1%). The types of tourism enterprises engaged in by the respondents were mainly scenic area (33.3%), guesthouse (26.6%), local specialty shop (9.2%), and agritainment (19.3%). A total of 93.7% of the respondents believed that the epidemic had affected their businesses to varying degrees. In addition, during the epidemic, more than half of the respondents believed that Guyan had given some help to residents during the epidemic, which laid the foundation for the follow-up questionnaire survey. The data is shown in Table 1.

From Table 2, it can be seen that the minimum KMO values for the measurement model as a whole were also within the acceptable range (0.751 > 0.7) and the significance values of the Bartlett’s tests were all 0.000, thus facilitating the next step of analysis. To assess potential common method bias, three analyses were conducted based on procedures described by Podsakoff (2003) as well as Kock (2015) [65,66]. First, the authors compared the goodness of fit between a single factor model using all the items with a multi-factor model whereby all the items were loaded as theorized. The results suggested that the multi-factor model yielded a significantly better fit. Second, the authors subjected all the items to a principal component factor analysis. This analysis yielded four factors, and the largest factor was at 28.85% variance, while none of the factors explained more than 44% of the variance. Third, the authors also assessed the variance inflation factor (VIF) scores for each construct. The VIF scores ranged from 1.279 to 1.713, much lower than the recommended value of 5 or lower to show the non-significance of multi-collinearity [66].

### 4.2. Measurement Model Evaluation

The minimum KMO values for the measurement model as a whole were also within the acceptable range (0.643 > 0.6) and the significance values of the Bartlett’s tests were all 0.000 (Table 2), thus allowing for the next step of the analysis. Next, the overall measurement model was evaluated through a two-step structural equation-modeling (SEM) analysis using primarily maximum likelihood estimation in SPSS AMOS 24, as suggested by Anderson and Gerbing (1988) [67], via a validated factor analysis (CFA) and an evaluation of the structural relationships to test the hypotheses. In terms of the overall model fit, the value of χ^2^/Df was 1.765 (*p* < 0.001, df = 129), the comparative fit index (CFI) = 0.949, Tucker–Lewis index (TLI) = 0.940, the root mean square error of approximation (RMSEA) was 0.061, and the normed fit index (NFI) = 0.892, which was an ideal fit. In addition, as shown in Table 3, the factor loadings of all the measurement question items ranged from 0.547 to 0.970. In summary, the overall measurement model demonstrated good reliability and enabled the next step of the hypotheses’ testing.

For the evaluation of convergent validity measured by the average variance extracted (AVE) (Table 3), the AVE values were higher than 0.5, except for the psychological well-being variable, which had an AVE value of 0.476, i.e., close to 0.5, which was also considered acceptable, indicating that most of the variance of each construct was explained by the adopted measurement items. Therefore, the reliability and convergence of the four variables are good. In addition, the AVE values of each variable are higher than the respective inter-variate correlations, indicating good discriminant validity [68]. In addition, the combined reliability values (CR) of the models were above the requirement of 0.70, showing good convergent validity [69]. In terms of discriminant validity, the square roots of the AVE values for all variables were greater than their respective values (Table 4). The relatively low AVE for psychological well-being may be attributed to the use of reverse scoring for the measurement questions at the time of the pilot study and data collection, which was a relatively new format for the respondents, rather than a discriminant validity issue. With an adequate measurement model fit achieved, the following structural relationship analyses were conducted.

### 4.3. Structural Model Testing

In this study, the Bootstrapping procedure was tested by SPSS AMOS24, and the model was run 2000 times within the 95% confidence interval using the great likelihood method to obtain the upper and lower bounds of the bias-corrected 95% confidence intervals. The results are shown in Table 5 and Table 6.

In the bivariate model of perceived community support and psychological well-being, the bias-corrected confidence interval does not contain 0 and the *p*-value is less than 0.05; thus, hypothesis 1 is supported. However, after adding the resident–tourist interaction and psychological resilience factors, the direct effects of perceived community support on psychological well-being were not significant.

A mediating effect is generally considered to exist when the bias-corrected confidence interval does not contain 0 and the *p*-value is less than 0.05. All the indirect effects of perceived community support on psychological well-being are significant. This means that H2 to H4 have been supported. Thus, psychological resilience positively mediates the relationship between perceived community support and psychological well-being (H2: a1 × b1). Likewise, the resident–tourist interaction mediates the path between perceived community support and psychological well-being (H3: a2 × b2). Finally, perceived community support is also positively associated with higher levels of psychological resilience and more resident–tourist interactions, which relates to higher levels of psychological well-being (H4: a1 × a3 × b2).

Since the direct effects of perceived community support and psychological well-being were not significant, all the postulated indirect effects are significant. Consequently, this means that psychological resilience and resident–tourist interactions fully mediate the relationship between perceived community support and psychological well-being. This is also supported by applying the variance-accounted-for (VAF) index. When the VAF has an outcome above 80%, a full mediation can be assumed [70].

Finally, we will test whether psychological resilience (M1) has a stronger mediating effect than resident–tourist interactions (M2). As in the previous example, we evaluate the statistical difference between a1 × b1, a2 × b2 and a1 × a3 × b2 (Table 6). In this case, we can observe a significant difference between both indirect effects. Compared with M2, M1 is more significant. In addition, the chain-mediating effect is significantly larger than the other two mediating effects.

## 5. Discussion and Conclusions

### 5.1. Discussion

To study the relationship between perceived community support and residents’ psychological well-being, we constructed a chain-mediating effect model based on organizational support theory, focusing on the important role played by two variables—namely, psychological resilience and resident–tourist interaction—in the relationship between perceived community support and residents’ psychological well-being. Local residents in the scenic area of Guyan, Lishui City, Zhejiang Province, were mainly selected for the survey. Based on the post-processing and analysis of the recovered questionnaire data, the following conclusions were drawn: (1) perceived community support is very helpful for the psychological well-being of residents; (2) there is a significant fully mediating effect of psychological resilience between perceived community support and residents’ psychological well-being interaction; (3) there is a significant fully mediating effect of resident–tourist interactions between perceived community support and residents’ psychological well-being; (4) after adding psychological resilience and resident–tourist interaction, the two variables played an orderly chain-mediating role between perceived community support and residents’ psychological well-being.

The theoretical model constructed in and findings of this study have theoretical implications for tourist destination residents’ psychological well-being and regional tourism development. 

First, the present study enriches the previous research by revealing the mechanisms underlying the relationship between perceived community support and psychological well-being. Previous studies have researched and confirmed the strong association of community involvement with health, well-being, and quality of life [71,72]. Most studies in the field of psychological well-being have examined some associations with variables such as stress, mood, work performance, and psychological capital [59,73,74]. Fewer studies have explored the impact of community support on residents’ psychological well-being in the tourism field. This paper creatively introduces two variables and reveals that perceived community support has a positive effect on both psychological resilience and resident–tourist interactions, and that resident–tourist interactions and psychological resilience can also contribute to psychological well-being. This result adds to the richness and comprehensiveness of research in the field of psychological well-being and demonstrates that psychological resilience and resident–tourist interactions are important means to connect perceived community support with the psychological well-being of residents in tourism destination communities.

Second, we introduced organizational support theory into the study of the relationship between community support and psychological well-being. The model constructed based on this theory and its findings enable special organizational structures—such as tourism destination communities—to be brought into focus, and thus enrich and expand the related research in the field of organizational support. Previous research in the area of organizational support mainly focuses on pay satisfaction, turnover intention, and job innovation of employees at the managerial level [75,76], and less attention is paid to the perception of support from the organization by members within the tourism destination community organizations. Thus, there is room for further research in the area of organizational support.

Finally, this study also highlights the important role of psychological resilience and resident–tourist interactions in the relationship between perceived community support and residents’ psychological well-being. The mechanisms of mutual influence were verified through multiple pathways, broadening the boundaries of the application of organizational support theory, and providing a new theoretical perspective for exploring the relationship between perceived community support and the psychological well-being of residents in tourism destinations.

### 5.2. Conclusions

This study examines the link between community support and psychological well-being in a population of residents in a tourist destination during the post-pandemic period. First, this study is guided by organizational support theory, which confirms the positive contribution of perceived community support to psychological well-being. Second, this study introduces the concept of psychological resilience and argues that psychological resilience is the key to residents’ recovery from a crisis and that higher resilience contributes to an increased probability of experiencing psychological well-being. The resident–tourist interaction is also taken into consideration, as residents who are willing to engage in friendly interactions with visitors are able to achieve better psychological well-being. In addition, resident–tourist interactions can have a synergistic effect with psychological resilience, which together contribute to the residents’ psychological well-being.

Based on the above findings, the authors hope that community managers in tourist destinations will pay attention to this issue in the post-pandemic period, considering that the current pandemic conditions may persist for an even longer period. As such, the authors propose several approaches that can improve these residents’ psychological well-being to enhance the practical implications of the study based on the literature supported by various organizations.

First, managers should aim to strengthen residents’ perceptions of community support. For tourist destination’s community management, such support can be enhanced by providing both tangible material support and intangible spiritual support to residents. Furthermore, in the post-pandemic era, heightened attention should be paid to residents’ emotions to prevent the spread of negative emotions. Second, managers should attempt to create a relaxed working atmosphere for residents to welcome the reopening of the tourism destinations. Sheldon and King (2001) argue that keeping employees in a good mood can create enhanced results for the company they work for [77]. By doing so, employees are able to engage in new tasks and enhance the frequency and quality of visitor interactions. Finally, a focus on self-care and self-improvement is warranted. Community residents should focus on improving their resistance to stress in the face of unknown risks, improving their psychological resilience, and promoting their psychological well-being. They should learn to shift their focus from themselves to tourists, take the initiative to communicate and interact with the tourists they meet, and actively share local characteristics with them to enhance the emotional experience of both parties. 

### 5.3. Limitations

There are some shortcomings in this study. First, the location of this study was the Guyan Scenic Area in Lishui City, Zhejiang Province, China, which is somewhat representative but still relatively narrow in scope. In further studies, the geographic and spatial scope can be expanded to include different areas with relatively large variability for a comprehensive comparative analysis to obtain more generalizable results. For example, Hyun et al. (2022) illustrated a significant association between social support and depressive symptoms during old age in both cross-sectional and longitudinal terms [78]. Second, the research design used in this application is cross-sectional, so it is difficult to draw more accurate causal conclusions, and subsequent studies can explore whether there is some variation in their results over time. Finally, this study only considered the mediating role of psychological resilience and resident–tourist interactions, and did not discuss whether there were other mediating variables or whether these two variables could, alone, act as intervening mechanisms to moderate the effect. In addition, traditional communities were chosen for this study, but in practice, online communities will attract more people because of the emergence of the COVID-19 pandemic. Russell et al. (2022) verified the relevant impact of the social relationships between healthcare personnel and online community members [79]. In addition, community members are considered to be part of the value chain, and many scholars have studied the influencing mechanisms of social support, belongingness, and value co-creation in online communities [80]. Therefore, more rich and interesting variables can be introduced and discussed in the future to better enrich the theoretical research framework of this paper.

## Figures and Tables

**Figure 1 ijerph-19-14553-f001:**
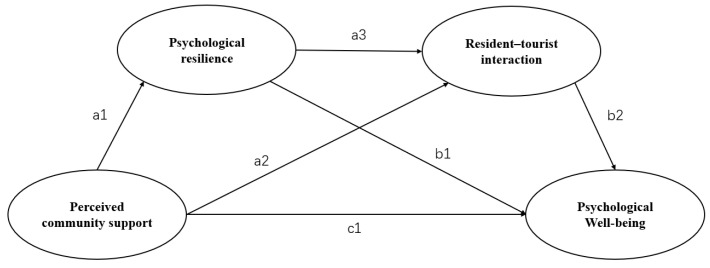
Conceptual framework.

**Table 1 ijerph-19-14553-t001:** Demographic information for participants.

Variable	Category	Count	Percentag (%)
Gender	Male	87	42.0
	Female	120	58.0
Age (years)	Age 18 and under	0	0.0
	19–30	13	6.3
	31–40	66	31.9
	41–50	71	34.3
	51–60	57	27.5
	Age 61 and older	0	0.0
Marital status	Married	200	96.6
	Unmarried	7	3.4
Education background	Under junior middle school	73	35.3
	High school degree	56	27.1
	Bachelor’s degree or above	78	37.7
Monthly income	Under RMB 2000	2	1.0
	RMB 2001~5000	106	51.2
	RMB 5001~8000	97	46.9
	RMB 8001~10,000	1	0.5
	Above 10,001	1	0.5
Type of business	Agritainment	40	19.3
	Guesthouse	55	26.6
	Local specialty shop	19	9.2
	Scenic area	69	33.3
	Others	24	11.6
The degree of impact on your career	No affect	2	1.0
	Neutrality	11	5.3
	Influenced	90	43.4
	Very influenced	49	23.7
	Highly influenced	55	26.6
Whether participant was permanent resident of Guyan	Yes	204	98.6
	No	3	1.4
Whether the community helped residents during the outbreak	Yes	107	51.7
	No	100	48.3

**Table 2 ijerph-19-14553-t002:** Reliability, KMO, and Bartlett test of sphericity.

		1	2	3	4	Total
Cronbach’s alpha coefficient		0.882	0.849	0.860	0.772	0.910
KMO		0.793	0.864	0.833	0.751	0.901
Bartlett’s test	chi-square	594.277	386.492	463.682	209.220	2030.909
	Df	6	10	10	6	153
	Statistical significance	0.000	0.000	0.000	0.000	0.000

Note: KMO is used to compare simple correlation coefficients between variables. 1 refers to perceived community support; 2 refers to psychological resilience; 3 refers to resident–tourist interaction; 4 refers to psychological well-being.

**Table 3 ijerph-19-14553-t003:** Assessment of the measurement model.

Construct	Items	Loading
Perceived community support		
	In the development of Guyan, the community will consider my views.	0.831
	In the development of Guyan, the community will look after my welfare.	0.825
	In the development of Guyan, the community will consider my personal ideas and goals.	0.801
	When I am in trouble, the community of Guyan will help me.	0.830
Psychological resilience		
	I can adapt to change.	0.816
	I can see the positive in a situation.	0.813
	Dealing with stress makes me a stronger person.	0.810
	I can face difficulties and work hard to achieve my goals.	0.822
	I can stay focused under pressure.	0.827
Resident–tourist interaction		
	I used to form friendships with tourists.	0.813
	My interactions with visitors were positive and helpful.	0.829
	I like interacting with tourists.	0.803
	I like to go to the tourist areas.	0.832
	I like to learn the culture of the tourists’ hometown.	0.842
Psychological well-being		
	I am responsible for my current life situation.	0.725
	I will take responsibility in life.	0.711
	I think it is important to have new experiences in my life that will challenge my view of myself and the world.	0.682
	For me, life is a continuous process of learning, changing, and growing.	0.718
Goodness-of-fit Indices	χ^2^/DF = 1.765, RMSEA = 0.061, CFI = 0.949, TLI = 0.940, NFI = 0.892, IFI = 0.950	

Note: χ^2^/DF refers to the ratio of chi-square to degrees of freedom. IFI refers to incremental fit index.

**Table 4 ijerph-19-14553-t004:** Testing discriminant validity.

	CR	AVE	1	2	3	4
Perceived community support	0.895	0.690	**0.831**			
Psychological resilience	0.849	0.530	0.426	**0.728**		
Resident–tourist interaction	0.863	0.559	0.464	0.719	**0.748**	
Psychological well-being	0.765	0.451	0.343	0.730	0.641	**0.671**

Note: CR refers to composite reliability. AVE refers to average variance extracted. The square roots of each corresponding AVE score are shown in bold.

**Table 5 ijerph-19-14553-t005:** Significance of direct and indirect effects.

Path	Coefficient		Bootstrap 95%				
			Lower			Upper	
Total effect	0.343		0.172			0.519	
Direct effects							
a1	0.426		0.254			0.570	
a2	0.192		0.042			0.326	
a3	0.637		0.507			0.764	
b1	0.557		0.340			0.803	
b2	0.244		0.022			0.307	
c1	−0.008		−0.154			0.148	
**Indirect Effects**	**Estimate**	**Lower 95%**		**Upper 95%**	** *p* **		**VAF**
a1 × b1	0.238	0.106		0.454	0.001		12.4%
a2 × b2	0.047	0.002		0.155	0.036		2.5%
a1 × a3 × b2	0.611	0.293		0.906	0.001		84.2%

Note: a1 × b1 refers to perceived community support–psychological resilience–psychological well-being; a2 × b2 refers to perceived community support–resident–tourist interaction–psychological well-being; a1 × a3 × b2 refers to perceived community support–psychological resilience–resident–tourist interaction–psychological well-being; c1 refers to perceived community support–psychological well-being.

**Table 6 ijerph-19-14553-t006:** Comparison of mediating effects.

	Estimate	Lower 95%	Upper 95%	*p*
Different effects				
M1 − M2	0.191	0.020	0.425	0.028
M1 − M3	−0.373	−0.661	−0.118	0.001
M2 − M3	−0.564	−0.875	−0.234	0.001

Note: M1 − M2 refers to (a1 × b1) − (a2 × b2); M1 − M3 refers to (a1 × b1) − (a1 × a3 × b2); M2 − M3 refers to (a2 × b2) − (a1 × a3 × b2).

## Data Availability

The data analyzed in this paper are proprietary, and, therefore, cannot be posted online.

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
