# Peer review of "Exploring the Relationship between Perceived Community Support and Psychological Well-Being of Tourist Destinations Residents"

_ijerph, 2022, doi:10.3390/ijerph192114553_

Round 1

Reviewer 1 Report

Dear authors,

The authors conducted a study to explore how perceived community support leads to psychological well-being. For that, the study explores two mediating mechanisms, namely psychological resilience and resident – tourist interaction.

Despite the positive features of the study, there are some considerations to take care of before the paper is published. 

The following comments will summarize my appreciation and major concerns with your paper. I hope these comments help you further improve your study.

Abstract: The abstract is confusing. Please re-structure it in a way clearer for the reader. Identify the main aim of the study, and the method used, as well as its implications, and limitations. 

Introduction:

1)    The introduction is confusing and longer. It would be useful that the authors to summarize the introduction and make a better connection between constructs.

2)    Specify the main aims of the study and the main contributions to the literature.

Theoretical background

1.     What are the main goals of the paper? What do you want to achieve? And what do you add to the literature?

2.     What theory may support your goals?

Method

3.     Please add more information regarding the data collection procedure. How did the participants were recruited? Was there any inclusion or exclusion criterion?

4.     In the measures section please:

a.     Add an item example for each measure. 

b.     If you use a seven-point likert scale, the scales ranged from 0 to 7 or from 1 to 7? Because in the first case you have an 8-point likert scale. Please clarify. 

5.     Please develop the data analysis procedure.

Discussion

 The discussion was short and did little more than summarise your findings. 

6.     Please, develop the discussion section, considering the theoretical implications.

7.     The limitations and future research should be elaborated.

8.     What are the main practical implications of the study? And what do you add that is not tested before?

Reviewer 2 Report

See details in the file attached

Round 2

Reviewer 1 Report

The authors did good work in addressing all my previous comments. I wish the authors good luck with their research.

Reviewer 2 Report

See the attached
